# Non-Melanoma Skin Cancer Detection in the Age of Advanced Technology: A Review

**DOI:** 10.3390/cancers15123094

**Published:** 2023-06-07

**Authors:** Haleigh Stafford, Jane Buell, Elizabeth Chiang, Uma Ramesh, Michael Migden, Priyadharsini Nagarajan, Moran Amit, Dan Yaniv

**Affiliations:** 1School of Medicine, Baylor College of Medicine, Houston, TX 77030, USA; 2Division of Internal Medicine, Department of Dermatology, The University of Texas MD Anderson Cancer Center, Houston, TX 77030, USA; 3Department of Pathology, The University of Texas MD Anderson Cancer Center, Houston, TX 77030, USA; 4Graduate School of Biomedical Sciences, The University of Texas, Houston, TX 77030, USA; mamit@mdanderson.org; 5Department of Head and Neck Surgery, The University of Texas MD Anderson Cancer Center, Houston, TX 77030, USA; dyaniv@mdanderson.org

**Keywords:** artificial intelligence, machine learning, deep learning, deep neural networks, non-melanoma skin cancer, basal cell carcinoma, squamous cell carcinoma, skin cancer screening, smartphone applications

## Abstract

**Simple Summary:**

Non-melanoma skin cancer is one of the most common cancer diagnoses in the world, and cases are rising globally. Consequently, novel technology to aid in screening is of interest. Deep learning is one type of artificial intelligence that has shown promise in image analysis and is an attractive tool for application in non-melanoma skin cancer diagnosis. Here, we review the evidence for deep learning in non-melanoma skin cancer screening and diagnosis, the available technologies and remote applications, and attitudes and ethical considerations regarding such technologies.

**Abstract:**

Skin cancer is the most common cancer diagnosis in the United States, with approximately one in five Americans expected to be diagnosed within their lifetime. Non-melanoma skin cancer is the most prevalent type of skin cancer, and as cases rise globally, physicians need reliable tools for early detection. Artificial intelligence has gained substantial interest as a decision support tool in medicine, particularly in image analysis, where deep learning has proven to be an effective tool. Because specialties such as dermatology rely primarily on visual diagnoses, deep learning could have many diagnostic applications, including the diagnosis of skin cancer. Furthermore, with the advancement of mobile smartphones and their increasingly powerful cameras, deep learning technology could also be utilized in remote skin cancer screening applications. Ultimately, the available data for the detection and diagnosis of skin cancer using deep learning technology are promising, revealing sensitivity and specificity that are not inferior to those of trained dermatologists. Work is still needed to increase the clinical use of AI-based tools, but based on the current data and the attitudes of patients and physicians, deep learning technology could be used effectively as a clinical decision-making tool in collaboration with physicians to improve diagnostic efficiency and accuracy.

## 1. Introduction

Skin cancer is the most commonly diagnosed cancer in the United Sates [1]. According to the current estimates, approximately one in five Americans will develop skin cancer during their lifetime, with non-melanoma skin cancer (NMSC) being the most prevalent form, and the incidence is increasing globally [2,3]. Consequently, methods are needed to reliably identify skin cancer on a larger scale so that lesions can be diagnosed earlier and necessary treatments can be performed in a timely manner, as untreated lesions can result in significant morbidity. Artificial intelligence (AI) has generated considerable interest in several fields of medicine, especially as a decision support tool to aid physicians in making clinical diagnoses [4,5,6]. While there are many types of AI, deep learning is one form that has shown to be particularly effective in image analysis [7]. Due to the visual nature of diagnoses in specialties such as pathology, radiology, and dermatology, deep learning could be diagnostically important across multiple disciplines. For example, the application of deep learning models to skin cancer screening and diagnosis is an attractive technique.

Ultimately, deep learning technology could be utilized to aid in skin cancer diagnosis in many ways. The widespread implementation of telemedicine following the COVID-19 pandemic has highlighted the feasibility and value of clinicians interacting with patients virtually [8], and the development of deep-learning-based telemedicine screening tools could further advance this ever-evolving patient–doctor relationship. Eventually, these tools could allow for easier access to medical services for the roughly 15 million Americans who live in medically underserved areas [9]. Furthermore, over the past two decades, mobile smartphones have advanced rapidly, with increasingly powerful cameras and operating processing power, creating the potential for deep-learning-based smartphone applications for skin cancer screening for remote patients. This review details the current state of deep learning technology used in NMSC diagnosis, the potential for smartphone applications to aid in screening, attitudes surrounding such technology, and important ethical considerations.

## 2. NMSC

NMSC is the most common form of skin cancer, and generally speaking, it is slow-growing and infrequently metastatic [3,10]. However, even without metastasis, there is still significant potential for morbidity and deformity depending on the lesion’s location. For example, NMSC of the head and neck can have a particularly significant impact on patients’ self-image and functional status, and adequate removal has been shown to improve patients’ body image perception [11]. For these reasons, early detection is critical for preserving a patient’s quality of life and preventing unnecessary morbidity. Overall, approximately 75% of NMSC diagnoses are basal cell carcinoma (BCC) [10]. BCC is a slow-growing lesion with a variety of initial presentations, but common clinical pictures include a shiny, flesh-colored papule with telangiectasias or a non-healing open sore [12]. Cutaneous squamous cell carcinoma (cSCC) is a less common form of NMSC that can also present in many ways, but it frequently presents as a scaling erythematous patch or plaque [13]. Dermoscopy is commonly used by dermatologists to better visualize and diagnose these skin cancers, where a handheld dermatoscope is used in a manner similar to a magnifying lens but with increased magnification and an adjustable illuminating system [14]. Ultimately, however, the varied presentations of NMSC have long presented a diagnostic challenge for clinicians, especially primary care physicians without extensive dermatology training, who are often a patient’s first and most accessible resource for skin concerns [15].

To this end, one retrospective review of medical records from a rural primary care clinic in Florida found that visits regarding skin cancer concerns were some of the most common dermatology-related encounters [16]. However, only 20% of patients were documented as having followed-up on their subsequent referral to a dermatologist. While several factors likely contribute to this situation, physical distance from providers may play a role, as the average distance to a dermatology referral center was 21 miles. This study highlights the need for easier access to skin cancer screening, particularly for rural patients who are physically distant from specialist-level care. Furthermore, the incidence of NMSC is increasing globally, with estimated increases in BCC of 145% and cSCC of 263% between 1976–1984 and 2000–2010 [17], further emphasizing the need for increased detection. If effective, AI-based technology could increase healthcare access and improve patient outcomes by streamlining the screening process—particularly for patients who have difficulty accessing care, such as rural Americans or underinsured patients.

## 3. Deep Learning Technology in NMSC Diagnosis

AI encompasses a wide range of technologies, and there are several subtypes. Machine learning is one type of AI that has many practical modern-day uses, such as speech transcription and object identification [7]. Machine learning structures are commonly based on supervised learning, in which an algorithm is given a training dataset to learn a known outcome—for example, to identify a skin lesion as squamous cell carcinoma. Then, data scientists can evaluate the accuracy of the machine learning model using validation data and testing data (Figure 1) [18]. This structure allows the program to follow learned patterns, but it is limited in its ability to gather complex data from an image in order to form new conclusions. Deep learning is a subset of machine learning based on deep neural networks (Figure 2) [19]. Conceptually, a deep neural network can be understood as a multi-layered algorithm in which the output of one layer becomes the input of the next. Subsequently, each layer of the network can form increasingly more complex conclusions. Once this system is in place, there is little need for human intervention.

Many studies have shown the utility of deep learning in several fields, including dermatology [20]. For example, in 2020, one study presented a deep learning system for establishing a differential diagnosis of skin conditions, with inputs based on skin photographs, various demographic information, and medical history [21]. This system achieved an accuracy of 66% when identifying the most likely diagnosis, compared to 66% for dermatologists, 44% for primary care providers, and 40% for nurse practitioners. This accuracy further increased when the algorithm was asked to identify the three most likely diagnoses. To reliably utilize deep learning technology for screening, the diagnostic sensitivity should be at least equivalent to that of trained experts in the field. This study demonstrated non-inferiority to specialists and superiority to non-specialists, showing the potential of deep learning technology to increase access to specialist-level diagnostics. In theory, this technology could be used to augment the workflow of specialists so as to help them to triage and diagnose cases more rapidly, enabling the specialists to focus more on lesions with a differential diagnosis including malignancy, for example.

With respect to skin cancer diagnosis, specifically, several studies have explored deep learning technologies for the detection of melanoma and showed non-inferiority [22,23] or even superiority [24] to specialists. Although fewer studies have investigated deep learning as a tool for NMSC detection, some studies have created deep learning frameworks to detect NMSC [25,26,27]. One quantitative review evaluated the success of deep learning technology in NMSC diagnosis and found no significant difference in the sensitivity or specificity of NMSC diagnosis between specialists using dermoscopy and machine learning techniques [28]. Once again, this finding demonstrates specialist-level sensitivity in NMSC diagnosis using machine learning, further emphasizing its potential utility in screening.

While dermoscopy has great utility in making a presumptive diagnosis of NMSC, the dermatopathology of biopsy specimens is ultimately considered to be the gold standard for NMSC diagnosis [29]. Recent research has demonstrated the potential of deep learning technology in the analysis of histopathological specimens, yielding high diagnostic accuracies for NMSC detection and classification [30,31,32]. Collectively, these findings highlight the potential of deep-learning-based technology to significantly transform NMSC screening and diagnosis—from dermoscopic analysis in the clinic to the pathological evaluation of biopsies in the lab.

Given the potential for a misdiagnosis of NMSC to have devastating clinical consequences for patients, the potential weaknesses of deep learning technology should be thoroughly explored and understood. To this end, one study investigated potential image perturbations that could cause a deep learning system to misidentify malignant melanoma as a benign nevus [33]. It was found that even small changes in color balance could impact the diagnosis of melanoma, where subtracting just 10 units from the green channel increased false negatives by 235%. Furthermore, it was found that rotating images by 45° and 180° each increased false negatives by 11%, despite the fact images were rotated randomly during the training of the network used. Notably, there was no significant difference in accuracy for dermatologists when presented with these altered vs. unaltered images.

Furthermore, another similar study found that skin markings using blue surgical ink markers led to an increase in false positives of approximately 40% [34]. Although these aforementioned studies primarily focused on the misdiagnosis of melanoma, it is reasonable to conclude that these findings would be similarly applicable to deep learning algorithms designed for the diagnosis of NMSC, given that any image of a suspicious skin lesion could be subject to these alterations. Ultimately, these studies highlight a common fault of deep-learning-based models, which is their generalizability [18]. Simply, this refers to the ability of an AI-based model to perform well when analyzing new data, and to avoid this, training datasets should be robust, diverse, and representative of the real-world scenarios the model will be utilized for. Moreover, such weaknesses should be identified with rigorous review prior to widespread clinical use to avoid any potential patient harm.

Although there are promising data regarding the use of deep learning and deep neural networks in skin cancer detection and diagnosis, there is still a long way to go before this technology can be widely implemented. Currently, there are few recognized applications of AI-based technology in clinical practice. However, one of the most prominent examples of this technology is the IDx-DR system for the autonomous detection of diabetic retinopathy from images, which was approved by the United States Food and Drug Administration in 2018 [35]. The device was initially reviewed through the FDA’s de novo pre-market review process, and it was granted designation as a “breakthrough device” after data from a study of 900 patients revealed that the system could correctly identify whether or not a patient had diabetic retinopathy almost 90% of the time [36]. This was the first such approval of AI-based technology in any medical field. As of October 2022, the FDA has approved more than 500 AI-enabled medical devices, with the majority of these devices (approximately 75%) being designated for use in radiology [37]. Notably, there have been no approvals for devices geared toward dermatologists. As research advances, it is reasonable to conclude that the FDA will continue to approve novel AI-based medical devices when there is adequate evidence for safety and efficacy.

## 4. Evidence for Remote Applications of Deep Learning Technology

Drawing on the promising ability for deep-learning-based technology to aid physicians in the diagnosis of NSMC, smartphone applications that are designed to screen for NMSC based on inputs from users’ smartphone cameras are an equally attractive possibility. Such an application could either serve as a direct-to-patient screening tool or function as a telemedicine tool that allows patients to forward images of their skin lesions to a dermatologist using built-in deep learning technology to screen and triage images. Here, we will primarily focus on the use of smartphone applications as a direct-to-patient screening tool. Few studies have examined the use of smartphone applications for diagnosing melanoma, ultimately revealing wide variability in the sensitivity and specificity of detection [38,39]. In one study comparing the performance of four smartphone applications, the application that sent images to a board-certified dermatologist for review had the highest sensitivity, and the other applications that only utilized algorithms had a poorer performance [38].

Fewer studies have examined the utility of smartphone applications in aiding NMSC diagnosis. However, in 2022, one study of deep learning algorithms compared dermoscopic images to smartphone images for the diagnosis of NMSC [40]. Ultimately, dermoscopic images had a significantly higher accuracy (0.88 vs. 0.75, *p* < 0.005) and sensitivity (95.3% vs. 75.3%, *p* < 0.001) than smartphone images, suggesting that using smartphone images for telemedicine could decrease the sensitivity of NMSC diagnosis as compared to dermoscopy. This is most likely because a dermatoscope utilizes magnification with a transilluminating light source to better visualize skin lesions by preventing obstruction from skin surface reflections, and smartphone cameras will likely always have this limitation, which should be considered when developing such applications. However, this technology could still be of immense value in low-resource settings, where patients may have limited or no physical access to a dermatologist but have access to a smartphone with internet connection. Even though the diagnostic accuracy is lower when using smartphone images compared to dermoscopy, it could be argued that less-sensitive screening is ultimately preferrable to no screening.

Ultimately, there are several barriers to consider with respect to the utilization of smartphone applications for NMSC screening, with one important consideration being the scalability of this technology. For example, when patients submit their skin photos, will these images be stored in some sort of database for future reference or for quality improvement purposes? If so, where will they be stored, and how will applications ensure that servers are large enough to contain this amount of data? Furthermore, who will bear the burden of the cost for such an application? Will patients be required to pay for the application, or perhaps more interestingly, could health insurance companies contribute to covering costs? Finally, although these applications may be deep-learning-based, will dermatologists serve as a “check” of the deep learning algorithms for quality assurance? If so, how many dermatologists would be needed to meet the volume of photos assessed by the application? All of these questions must be carefully considered before any widespread application of this technology can be achieved.

## 5. Currently Available Applications

Several AI-based applications for skin cancer screening are currently available. Based on a search of the Apple application store, many of these applications are geared towards helping patients to identify and monitor concerning moles with malignant features. “Scanoma” is one such app; it uses AI-based technology to photograph moles, identifying them as likely benign or concerning, with the option to forward these images to a board-certified dermatologist for review. Other similar applications include “Skin-Check” and “Piel”, although these applications do not feature the option for patients to communicate with dermatologists. Given that these applications are primarily advertised to help patients to scan, track, and identify moles with malignant features, they may be of little utility in NMSC screening, given that BCC and cSCC do not characteristically present as moles.

Another example application is “Aysa”, a system built on resources from VisualDx, which generates a list of possible diagnoses based on a photo of a patient’s skin, their symptoms, and other demographic information. It also provides relevant information regarding the differential diagnosis. This application format may be more useful in helping patients to identify lesions that concerning in regard to NMSC, as it can be used for any type of lesion. Ultimately, this is not an exhaustive list of potential applications—a Google search of “phone app for skin cancer screening” reveals approximately 193,000,000 results, emphasizing the many potential tools and interest in this subject.

## 6. Attitudes Surrounding Deep Learning Technology

As outlined thus far, the potential for AI and deep learning technology to aid in NMSC diagnosis is promising; however, many researchers and patients still have reservations about its use in everyday practice. Understanding such attitudes is crucial for ensuring that common concerns can be addressed; ultimately, if physicians are hesitant to use AI-based technology or if patients feel uncomfortable with its use, these strategies may never translate into clinical practice.

In 2019, one global survey of pathologists attempted to better characterize clinicians’ beliefs regarding the use of AI in clinical practice [41]. Most respondents (58%) thought that AI-based tools could lead to increased diagnostic efficiency, and many (53%) felt that the use of such technology as a decision support tool could decrease the rates of reported errors. However, many respondents (48.3%) still felt that diagnostic decision making should remain primarily in the hands of clinicians. A more recent 2021 survey involving multiple medical specialties found that most dermatologists (54.6%) expected their practice to be affected by AI more than the practice of other medical specialties [42]. Within this cohort, the most cited concern was anxiety regarding medical liability in the context of machine error; however, improved access to disease screening was perceived to be the greatest benefit. Volunteer response bias should be considered when interpreting these surveys, and these views may not be generalizable.

Given the potential for AI to significantly transform healthcare access and engagement, patient perspectives regarding this technology are just as important to understand as those of clinicians. One recent patient survey sought to quantify attitudes regarding the use of AI in skin cancer screening, specifically [43], and patients’ perceived benefits included faster diagnostic speeds, increased healthcare access, and greater patient activation. Common concerns included less accurate diagnosis and increased patient anxiety. A staggering 94% of patients emphasized the importance of collaboration between artificial and human intelligence.

## 7. Ethical Considerations

Based on the aforementioned studies, physician supervision of AI-based skin cancer screening is a common theme that is viewed positively by both patients and providers. From a medical ethics perspective, it could be argued that without the clinical input of licensed physicians, the widespread use of direct-to-patient screening tools could lead to increasing moral hazards. For example, the sense of security offered by a smartphone app may empower patients to spend more time outside, wear less sunscreen, or visit a physician less often. This will be problematic if such technologies are any less sensitive in screening for skin cancer than physicians are. Health literacy and familiarity with technology may also play a role here, and it could be argued that patients with more reliable access to technology will be more likely to utilize such screening tools, further widening existing gaps in care access. Finally, the emotional impact of this type of technology should also be considered; the stress that an application could cause by telling a patient that their skin lesion is malignant should not be understated. Theoretically, physicians could inform patients of their skin cancer diagnosis, prognosis, and treatment with much more nuance, which could drastically influence a patient’s initial emotional reaction, helping to alleviate their specific fears.

Other important ethical considerations regarding the general use of artificial intelligence in medical decision making include the issues of patient privacy and algorithm transparency. Collecting and storing images for the purpose of making a diagnosis requires the handling of sensitive patient information; thus, AI-based technology should be held to strict privacy regulations. Furthermore, patients have the right to understand how their data are being used, in addition to the decision-making process utilized by the AI-technology. In order to ensure patient and provider trust in AI-based technologies, these processes and regulations should be made as transparent as possible.

Finally, it should not be overlooked that these technologies are still fallible, because as mentioned previously, machine-made decisions based on human-designed algorithms are subject to bias depending on the inputted data. When robust training data are not used, AI-based models may not be generalizable, which may result in patient harm. Unfortunately, this has the potential to further worsen already existing healthcare disparities. Demonstrating this point, a scoping review of dermatologic machine learning research found that, of 136 reviewed studies, only 12 clearly disclosed the race or ethnicity of image sources [44]. Other studies have shown a similar lack of transparency with respect to the distribution of race in datasets used to train deep learning algorithms [45]. Therefore, to avoid potentially exacerbating already-existing healthcare disparities in regard to patients of color, algorithms for skin cancer screening and detection should be developed using inclusive, large, and representative data that contain images of malignancy on all skin tones.

## 8. Ongoing Research and Future Directions

In addition to the data presented here, further research regarding the use of AI-based technology in NMSC screening is underway. Information regarding the ongoing or recently completed research is summarized in Table 1. Trials were found through a search of clinicaltrials.gov using the keywords “non-melanoma skin cancer” and “artificial intelligence”. Notably, only three projects were found, and this was confirmed with a broader search of the website including the term “non-melanoma skin cancer” alone. Two of these projects were designed specifically to investigate a particular AI-based diagnosis support tool, Deep Ensemble for the Recognition of Malignancy (DERM), and both studies were recently completed within the past two years. This technology was shown to be effective in identifying lesions that are suspicious for malignant melanoma [46], but data regarding its efficacy in NMSC diagnosis have not yet been published. The final trial is currently recruiting, and it was designed to investigate the effects of teledermoscopy in clinical practice, how to introduce AI-based technology within teledermoscopy visits, and if there are any changes to diagnostic accuracy with the implementation of this technology.

Overall, further research is needed regarding the implementation of AI-based diagnosis support tools in clinical practice. Evidence has shown that this technology can be effective, but it remains unclear whether this will result in appreciable changes in patient outcomes. Furthermore, there is little understanding of the cost-effectiveness of this kind of technology—both for patients and for the healthcare system overall. These are important barriers to overcome before AI-based technology can be widely implemented in clinical practice and patient care.

## 9. Conclusions

In the post-pandemic world, telemedicine is increasingly becoming a part of clinical practice. Furthermore, as cases of skin cancer are rising globally, telemedicine is an attractive avenue for NMSC diagnosis, and novel technologies to aid in screening are of interest. The available data for the detection and diagnosis of NMSC using deep learning technology are promising, revealing sensitivity and specificity that are not inferior to those of trained dermatologists. However, the translation of this technology to smartphones will likely be problematic owing to the decreased sensitivity and specificity associated with smartphone cameras, which lack the magnification and transilluminating light source of a dermatoscope.

There is still work to be done to increase the use of AI-based tools in clinical practice, as few are currently widely accepted. However, based on the information gathered here, including the attitudes of patients and physicians, deep learning technology could be effectively used as a clinical decision-making tool in collaboration with physicians to improve diagnostic efficiency and accuracy, particularly in populations with limited to no access to dermatologic resources.

## Figures and Tables

**Figure 1 cancers-15-03094-f001:**
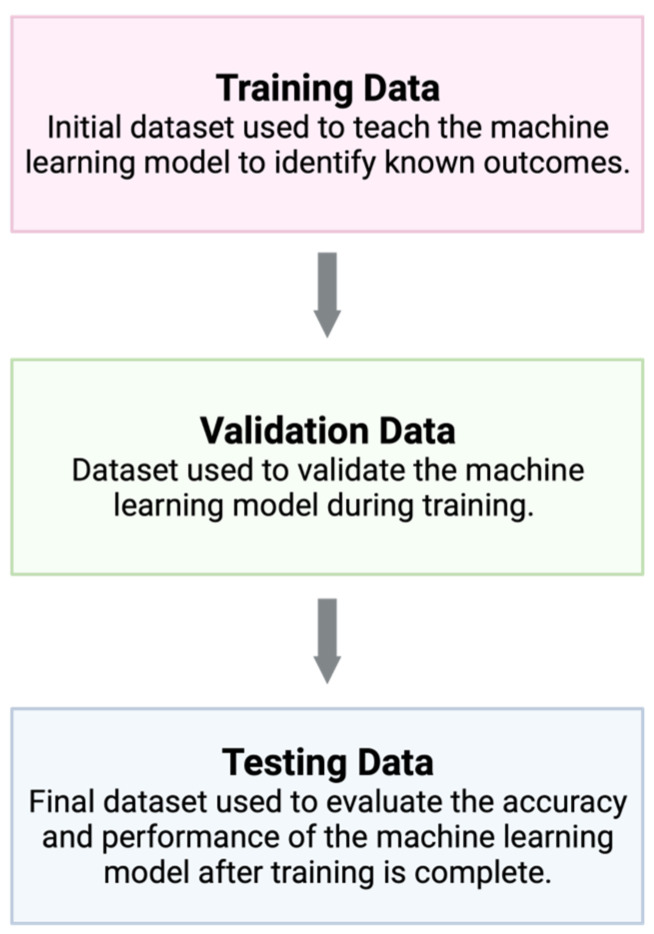
Machine learning models are based on supervised learning, where the model is first exposed to a training dataset in order to learn a given outcome. Validation data are used to frequently assess the machine learning model during training and development, and finally, testing data are used to evaluate the performance of the model once training is finished. Created with BioRender.com.

**Figure 2 cancers-15-03094-f002:**
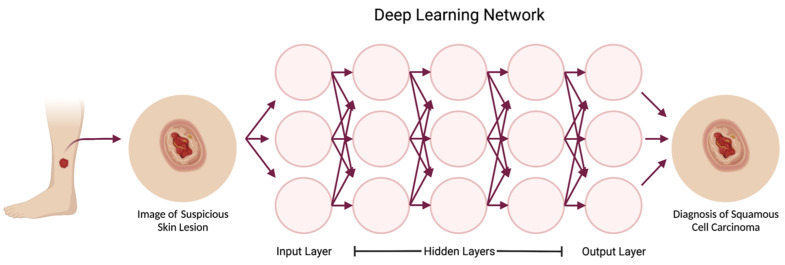
Deep learning is a subset of machine learning based on deep neural networks, where multiple sequentially ordered layers allow for the network to form increasingly complex conclusions. The input layer includes pixels from the photograph, while each hidden layer extracts certain features from the photo to reach a conclusion in the output layer. In this example, the first hidden layer might detect the edges of the lesion, and deeper hidden layers might recognize that certain edge patterns are consistent with a neoplasm. The deepest layer would classify the lesion into the output category of squamous cell carcinoma. Created with BioRender.com.

**Table 1 cancers-15-03094-t001:** Ongoing and recently completed research regarding the use of AI-based technology in NMSC diagnosis. Information was obtained from clinicaltrials.gov.

Study Name	Study Goal	Patient Population	Design	Clinicaltrials.gov Identifier
Effectiveness of an Image Analyzing Algorithm (DERM) to Diagnose Non-melanoma Skin Cancer (NMSC) and Benign Skin Lesions Compared to Gold Standard Clinical and Histological Diagnosis	To establish how well the device DERM (Deep Ensemble for the Recognition of Malignancy, an AI-based diagnosis support tool) determines the presence of non-melanoma skin cancer in images of skin lesions collected in a clinical setting.	Patients attending a dermatology clinic with at least one suspicious skin lesion.	Prospective cohort study (actual enrollment: 572 participants)	NCT04116983
Impact of an Artificial Intelligence Platform (DERM) on the Healthcare Resource Utilization (HRU) Needed to Diagnose Skin Cancer When Used as Part of a United Kingdom-based Teledermatology Service	To establish whether the use of DERM (Deep Ensemble for the Recognition of Malignancy) in the patient pathway could reduce the number of unnecessary referrals for dermatologist review and/or biopsy.	Adult patients undergoing medical photography for imaging of at least one suspicious skin lesion.	Prospective cohort study (actual enrollment: 700 participants)	NCT04123678
Teledermoscopy and Artificial Intelligence: Effects of Implementation in Clinical Practice	To investigate the effects of utilizing teledermoscopy in clinical practice, including changes in referral patterns and effects on diagnostic accuracy. It will also investigate how to introduce AI within teledermoscopy.	Patients aged 15 years or older with a skin lesion that is assessed by a physician who are subsequently referred for teledermoscopy.	Prospective cohort study (estimated enrollment: 8000 participants)	NCT05033678

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
