# Peer review of "Non-Melanoma Skin Cancer Detection in the Age of Advanced Technology: A Review"

_cancers, 2023, doi:10.3390/cancers15123094_

Round 1

Reviewer 1 Report

Dear Authors,

The review article by Stafford and the team on DL-based detection of NMSC is interesting and concise. However, the review manuscript could be improved.

Here are some suggestions:

1.     There is a plethora of DL-based models, however, they suffer from one common problem, which is generalizability. AI-based models are as good as the training data it is provided. Although the authors have already mentioned some limitations, the limitations of the DL-based model should be expanded to include all other possible limitations.

2.     Terms such as training data, testing data, and validation data could be confusing to some audiences. A schematic diagram of steps involved in DL based model would help readers to understand such terms.

3.     In lines 45-46, the Authors mention “Artificial intelligence (AI) has generated considerable interest in the field of medicine, especially as a decision-support tool to aid physicians in making clinical diagnoses.”  Such statements could be supported with references.

It is recommended that the author cite the following article which provides the most up-to-date information on FDA-approved AI/ML-based medical devices.

“https://www.medrxiv.org/content/10.1101/2022.12.07.22283216v2.full-text”

Minor.

Author Response

Response to Reviewer 1 Comments

Point 1: There are a plethora of DL-based models, however, they suffer from one common problem, which is generalizability. AI-based models are as good as the training data it is provided. Although the authors have already mentioned some limitations, the limitations of the DL-based model should be expanded to include all other possible limitations.

Response 1: This is a very good point. To address this important limitation, we have included a discussion of generalizability in lines 194-199 of the manuscript. In addition, we have expanded upon the idea as it relates to skin color in lines 413-424.

Point 2: Terms such as training data, testing data, and validation data could be confusing to some audiences. A schematic diagram of steps involved in DL based model would help readers to understand such terms.

Response 2: Thank you for this suggestion—we agree that these terms could be explained further to avoid confusion. Accordingly, we have included a new schematic diagram on page 2 of the manuscript to explain training data, validation data, and testing data as it relates to machine learning models.

Point 3: In lines 45-46, the Authors mention “Artificial intelligence (AI) has generated considerable interest in the field of medicine, especially as a decision-support tool to aid physicians in making clinical diagnoses.” Such statements could be supported with references.

Response 3: Thank you for bringing this to our attention. We have updated the manuscript with relevant references in support of this statement.

Point 4: It is recommended that the author cite the following article which provides the most up-to-date information on FDA-approved AI/ML-based medical devices:“https://www.medrxiv.org/content/10.1101/2022.12.07.22283216v2.full-text”

Response 4: Thank you for directing us to this article. We agree that is important to include within the manuscript, and the article has been cited in lines 209-226.

Reviewer 2 Report

I thank the academic editor for the opportunity to review this interesting manuscript on the use of artificial intelligence in the field of dermatology. I believe that as a review paper it is very well conducted and publishable in the prestigious journal cancers, but I would suggest a lesser revision for the following reasons:

1. I would ask the authors to add a paragraph on the use of artificial intelligence in Dermatopathology.

2. Check for typos.

minor revision also for english

Author Response

Response to Reviewer 2 Comments

Point 1: I would ask the authors to add a paragraph on the use of artificial intelligence in Dermatopathology.

Response 1: Thank you for this suggestion. In the original version of this manuscript, the subject of dermatopathology was mentioned in lines 135-142. Based on the reviewer’s comment, we have updated the manuscript to improve the clarity of this section for the reader, with the new paragraph found in lines 170-177. We hope this is sufficient, as the current literature in this area is limited.

Point 2: Check for typos.

Response 2: As suggested by the reviewer, we have carefully inspected the manuscript for incorrect spelling and grammatical errors, all of which have been corrected.